# Filamentous Temperature-Sensitive Z Protein J175 Regulates Maize Chloroplasts’ and Amyloplasts’ Division and Development

**DOI:** 10.3390/plants14142198

**Published:** 2025-07-16

**Authors:** Huayang Lv, Xuewu He, Hongyu Zhang, Dianyuan Cai, Zeting Mou, Xuerui He, Yangping Li, Hanmei Liu, Yinghong Liu, Yufeng Hu, Zhiming Zhang, Yubi Huang, Junjie Zhang

**Affiliations:** 1College of Life Science, Sichuan Agricultural University, Yaan 625014, China; lhyal2o3@163.com (H.L.); hxiaowu@stu.sicau.edu.cn (X.H.); m19180173072@163.com (H.Z.); d1anyuanc@163.com (D.C.); tingting20317@163.com (Z.M.); hxr511381@163.com (X.H.); hanmeil@163.com (H.L.); 2State Key Laboratory of Crop Gene Exploration and Utilization in Southwest China, Sichuan Agricultural University, Chengdu 611130, China; liyangping163@163.com (Y.L.); huyufeng@sohu.com (Y.H.); yubihuang@sohu.com (Y.H.); 3Maize Research Institute, Sichuan Agricultural University, Chengdu 611130, China; 13964@sicau.edu.cn; 4State Key Laboratory of Crop Biology, College of Life Sciences, Shandong Agricultural University, Taian 271018, China; zhzhang@sdau.edu.cn

**Keywords:** maize, chloroplast, amyloplast, FtsZ

## Abstract

Plastid division regulatory genes play a crucial role in the morphogenesis of chloroplasts and amyloplasts. Chloroplasts are the main sites for photosynthesis and metabolic reactions, while amyloplasts are the organelles responsible for forming and storing starch granules. The proper division of chloroplasts and amyloplasts is essential for plant growth and yield maintenance. Therefore, this study aimed to examine the *J175* (*FtsZ2-2*) gene, cloned from an ethyl methanesulphonate (EMS) mutant involved in chloroplast and amyloplast division in maize, through map-based cloning. We found that *J175* encodes a cell division protein, FtsZ (filamentous temperature-sensitive Z). The FtsZ family of proteins is widely distributed in plants and may be related to the division of chloroplasts and amyloplasts. The J175 protein is localized in plastids, and its gene is expressed across various tissues. From the seedling stage, the leaves of the *j175* mutant exhibited white stripes, while the division of chloroplasts was inhibited, leading to a significant increase in volume and a reduction in their number. Measurement of the photosynthetic rate showed a significant decrease in the photosynthetic efficiency of *j175*. Additionally, the division of amyloplasts in *j175* grains at different stages was impeded, resulting in irregular polygonal starch granules. RNA-seq analyses of leaves and kernels also showed that multiple genes affecting plastid division, such as *FtsZ1*, *ARC3*, *ARC6*, *PDV1-1*, *PDV2*, and *MinE1*, were significantly downregulated. This study demonstrates that the maize gene *j175* is essential for maintaining the division of chloroplasts and amyloplasts and ensuring normal plant growth, and provides an important gene resource for the molecular breeding of maize.

## 1. Introduction

Maize, a vital C4 plant, serves as a food crop and an important industrial raw material. Chloroplasts are the primary sites for photosynthesis and numerous metabolic reactions. Chloroplast division is crucial for the survival and reproduction of various plant species in nature [1,2]. The division of amyloplasts significantly influences the size, morphology, and yield of starch granules [3,4]. In cells with varying physiological and developmental states, chloroplasts and amyloplasts undergo different degrees of division. This process is governed by multiple pathways that require further refinement [5,6].

Chloroplasts are photosynthetic organelles in plants that convert energy. They have a double-layered membrane structure [7]. They are believed to have evolved from cyanobacteria and replicate through a process called binary fission [8]. The mechanism responsible for chloroplast division mainly involves four ring-like structures: the FtsZ ring (Z ring), inner and outer plastid-dividing rings, and the DRP5B (Dynamin-related proteins 5B) ring. These components contract and compress the division site to facilitate chloroplast division [7,8,9]. The Z ring, formed by chloroplast FtsZ proteins of cyanobacterial origin, is the first ring structure formed during chloroplast division [9]; it belongs to the GTPase family, which is associated with tubulin and acts as a scaffold protein for the division complex [1,5]. In plants, FtsZ proteins are composed of two subfamilies: FtsZ1 and FtsZ2. These proteins assemble into a polymer and form a ring at the cleavage site [10,11]. The FtsZ proteins in chloroplasts are anchored to the inner membrane through an interaction between the C-terminal domain of FtsZ2 and the N-terminal domain of the transmembrane protein of chloroplast accumulation and replication (ARC6) [7]. In *arc6* mutants, filamentous FtsZ proteins are fragmented, indicating that ARC6 is crucial for the assembly and stability of FtsZ rings [1]. The chloroplast outer-envelope membrane protein PDV2 and its paralog PDV1 recruit the cytosolic dynamin-related GTPase ARC5 (accumulation and replication of chloroplasts 5) to the chloroplast division site [12], which determines the rate of chloroplast division [13,14]. Studies have demonstrated that the size of chloroplasts can be altered through either the overexpression or antisense repression of *FtsZ* [15,16,17]. Overexpression or antisense repression of *AtFtsZ1* or *AtFtsZ2* in Arabidopsis resulted in the formation of one or few large chloroplasts per cell. A threefold increase in AtFtsZ1-1 protein levels inhibited chloroplast division [16,17]. Higher AtFtsZ1-1 protein levels resulted in more severe phenotypes. The defects in chloroplast division caused by AtFtsZ1-1 overproduction may reflect a stoichiometric imbalance among the components necessary for chloroplast division [17]. Furthermore, a decrease in StFtsZ1 protein levels in potato leaves resulted in a reduction in the number of chloroplasts in guard cells [18]. The role of the FtsZ protein in Z ring assembly during chloroplast division in maize requires further investigation.

The amyloplast reaction in cereals can coordinate with photosynthesis occurring in the leaves, allowing the products of both reactions to be mutually transformed under light induction [19,20]. Ultrastructural, molecular, and genetic data indicate that the components required for the division process are similar across all plastid types [5,17]. The mutant parc6, which affects the splitting of wheat chloroplasts and amyloplasts, exhibited enlarged plastids in leaves and endosperm. Additionally, the endosperm amyloplasts of this mutant contained a higher proportion of A-type and B-type starch particles compared to the wild-type [21]. An increase in StFtsZ1 protein levels results in a significant reduction in the number of starch granules and an increase in their size in tubers [18]. Further analysis is needed to determine whether FtsZ in maize affects grain starch splitting.

Therefore, this study identified, for the first time, the FtsZ (filamentous temperature-sensitive Z) family gene *j175* (*ZmFtsZ2-2*) that regulates chloroplast and amyloplast division in maize. During the seedling stage, white stripes began to emerge on the leaves of the *j175* mutant, accompanied by a significant decrease in photosynthesis. Electron microscopy observations revealed abnormal division of chloroplasts and amyloplasts in the mutant, resulting in a significantly lower grain weight compared to that of the wild-type. These research findings not only enhance our understanding of the molecular mechanism underlying chloroplast and amyloplast division in maize, but also provide valuable genetic resources for improving maize germplasm through molecular regulation.

## 2. Results

### 2.1. Leaf Phenotypic Identification of j175

The *j175* mutant was derived from EMS mutagenesis of the inbred line RP125 and was continuously backcrossed for purification. The leaves of the *j175* mutant seedlings were yellowed and grew weaker than those of the WT (Appendix A). During the jointing stage, the leaves changed from yellow to green. At this stage, clear white stripes emerged on the leaves of *j175* (Figure 1A)*,* and the *j175* plants exhibited a shorter stature compared to the WT plants (Figure 1C–E). The leaf length, width, and thickness of *j175* during the jointing stage were significantly lower than those of the WT (Appendix A).

### 2.2. Chloroplast Division Defect in j175

We observed that the number of chloroplasts decreased, while the volume increased, in the green parts of leaves of the *j175* mutant. Furthermore, the white parts of leaves of the *j175* mutant had almost no chloroplasts (Figure 1M–O). This suggests that the chloroplast division ability of the *j175* mutant is significantly reduced, particularly in the white parts of leaves. The chloroplasts of *j175* leaf mesophyll cells during the seedling and mature stages were much larger than those of the WT, indicating that this division defect accompanies the entire growth period of the plant (Figure 1K,L and Appendix A).

### 2.3. Chlorophyll Content of j175 Decreases

The photosynthetic capacity of leaves is closely related to their chlorophyll content. Almost no chlorophyll was detected in the white parts of leaves of *j175*. In contrast, the chlorophyll a, chlorophyll b, and total chlorophyll content in the green parts of leaves of *j175* were significantly reduced, with chlorophyll a and b decreasing by 15.43% and 27.95%, respectively (Figure 1B).

### 2.4. j175 Photosynthetic Rate Decreases

The net photosynthetic rate refers to the organic matter accumulated during photosynthesis in plants. The net photosynthetic rate of the green parts of leaves of *j175* exhibited a decrease of 22.26%, while the rate for the white parts of leaves decreased substantially, by 97.93% (Figure 1F). The transpiration rate refers to the capacity of plants to regulate water loss through leaf stomata as they adapt to their natural environment. The transpiration rates of the green and white parts of *j175* leaves exhibited a decrease of 26.50% and 94.60%, respectively (Figure 1G). The size of stomatal conductance indicates the extent of stomatal opening in plant leaves, which influences the ability of plants to absorb CO_2_ for photosynthesis. The stomatal conductance of the green and white parts of *j175* leaves decreased by 30.10% and 95.11%, respectively (Figure 1H). The intercellular CO_2_ concentration is inversely proportional to photosynthesis. Compared to the wild-type leaves, no significant change was observed in intercellular CO_2_ concentration in the green parts of j175 leaves. However, the intercellular CO_2_ concentration in the white leaf parts increased by 504.38% (Figure 1I). In summary, the mutant showed a significant decrease in the photosynthetic capacity of its green leaf parts, while photosynthetic capacity was nearly absent in the white leaf parts.

### 2.5. Determinations in Enzyme Activity of j175

The measurements for soluble sugar content, MDA content, and CAT activity showed no significant difference between the green leaf parts of *j175* and the WT. In the white leaf parts, the soluble sugar content, MOD content, and CAT activity decreased by 75.60%, 69.70%, and 61.80%, respectively (Appendix A). In contrast, the POD activity in the green and white leaf parts of *j175* decreased significantly, by 49.77% and 53.85% (Figure 1J), respectively. These results show a decrease in the ability of *j175* to resist abiotic stress.

### 2.6. Ear Phenotypic Identification of j175

The mature ears of *j175* were much shorter than those of the WT, and the kernels were significantly smaller than those of the WT (Figure 2A–C). Additionally, the height of the plant, ear height, hundred-kernel weight, kernel length, kernel width, and kernel thickness of *j175* were all reduced to varying degrees. However, no significant change was observed in the number of kernels per ear in *j175* (Figure 2D–L).

### 2.7. J175 Exhibits Abnormal Development of Starch Granules and Amyloplasts

Scanning electron microscopy was utilized to observe the endosperm starch in mature grains of the wild-type and *j175*. The WT starch granules exhibited smooth, spherical shapes arranged in a regular and orderly manner (Figure 2N). In contrast, the starch granules of *j175* existed in the form of oligomers, with a smooth surface and compact arrangement, and their number was significantly reduced (Figure 2O). The total starch content of endosperm cells was measured, and it was found that the total starch content of *j175* decreased by 3.85% (Figure 2M).

Amyloplasts and chloroplasts are types of plastids that share similarities in their division processes [17]. Endosperm cells were selected for observation using fluorescence microscopy. The WT endosperm amyloplasts at 12 DAP displayed a smooth spherical shape, whereas the mutant endosperm amyloplasts exhibited a typical bead-on-a-string shape during the powder-making division. However, some of the amyloplasts displayed relatively regular polygon shapes with smooth surfaces (Figure 2P,Q). At 20 DAP, the *j175* endosperm amyloplasts developed further into irregular polygons with multiple starch particles clustered together, significantly reducing the number of amyloplasts (Figure 2R,S). These results indicate that the mutation of the J175 gene affects amyloplast division in the grain, ultimately forming a complex of oligomers.

### 2.8. Positional Cloning and Allelic Testing Confirmed That J175 Encodes the FtsZ2-2 Protein

The homozygous *j175* mutant was crossed with the B73 inbred line, and F1 *j175*/+ plants were self-crossed to create the F2 mapping population. The chi-square test of the F2 population confirmed that the mutant phenotype was a recessive mutation controlled by a single gene (Appendix A). BSA-seq analysis, conducted on F2 wild-type and mutant segregant populations, identified the location of the gene near the Bin value of 10.07 on chromosome 10 (RefGen_v4) (Appendix A). Subsequently, using the surrounding insertion-deletion (In Del) markers from the 962 individuals in the F2 mapping population, the candidate interval was narrowed down to 144kb (Figure 3A). This region contains four coding genes. PCR amplification and sequencing identified a single-base mutation (G to A) in exon 4 of Zm00001d026669, resulting in an amino acid change from glycine (GGG) to arginine (AGG), which altered the function of the gene (Figure 3A,D). This functional change likely contributes to the emergence of the *j175* phenotype.

To determine the candidate gene Zm00001d026669, we conducted allele tests. The EMS mutants *j175-1* and *j175-2* of this gene were obtained from maizeEMSDB. The mutations in *j175-1* and *j175-2* are both C-to-T substitutions, occurring on exon 2 and exon 7, respectively, which results in the transformation of an arginine and asparagine residue to a premature stop codon, leading to the loss of some functional domains in Zm00001d026669 (Figure 3A,D). The leaves of self-pollinated homozygous *j175-1* and *j175-2* exhibit the same white-striped phenotype as *j175*, and their agronomic traits have also been measured to have varying degrees of reduction (Appendix A). The chloroplast division of the mesophyll cells of j175-1 and j175-2 was also inhibited, and the morphology of starch granules was also characterized by non-rough polygons, which was quite different from the smooth spherical morphology of the WT (Appendix A). *j175* was hybridized with *j175-1* and *j175-2*, and the offspring *j175/175-1* and *j175/j175-2* both exhibited a white-striped phenotype (Figure 3C). These results confirm that Zm00001d026669 is the causative gene for *j175*.

Genome sequencing revealed that the protein encoded by J175 consists of 467 amino acids and has a molecular weight of 57.67 kDa. In MaizeGDB, the Zm00001d026669 gene is annotated as a cell division protein FtsZ homolog [22]. In this study, Zm00001d026669 is referred to as J175. Protein sequence analysis indicated that Zm00001d026669 encodes FtsZ2-2 (J175), a homologous protein of the FtsZ family, which is highly conserved in evolution. Maize contains only three FtsZ proteins: FtsZ1, FtsZ2-1, and FtsZ2-2. FtsZ2-2 shares a high degree of sequence similarity (77.8%) with FtsZ2-1, indicating similar functions. J175 exhibits high sequence similarity (79.80%) to AtFtsZ2-2 (Figure 3B). AtFtsZ2-2 is involved in chloroplast division, indicating that J175 may play a role in the process of chloroplast division in maize [17]. Additionally, conservative domain analysis indicated that the missense mutation site (Gly to Arg) of *J175* is located at a highly conserved site in the FtsZ protein, with the mutated SNP occurring in the second motif (Appendix A). The glycine at this site is highly conserved. We speculate that the mutation at this site affects the function of the protein, leading to the inhibition of chloroplast division and potentially resulting in a complete loss of division ability.

### 2.9. Subcellular Localization and Constitutive Expression of j175

To determine the subcellular localization of J175, we fused J175 with GFP and expressed the fusion protein in maize leaf protoplasts. Furthermore, as a control, signals from 35S: GFP were detected in the nucleus and cytoplasm. In contrast, the signal of 35S: J175 GFP overlapped with the plastid marker (Figure 3E), suggesting that J175 functions within the plastid. Protoplasts were obtained from the leaves of etiolated seedlings after 2 weeks of germination. Photos were captured by laser scanning confocal microscopy.

The expression of the *J175* gene in different tissues of wild-type RP125 at the V13 stage (the time when the plant rapidly transitions from vegetative to reproductive growth) was examined using qRT-PCR. *J175* exhibited constitutive expression and was expressed across various tissues, with the highest expression levels in the bracts and female tassels, and the lowest in the male tassels (Figure 3F). Analysis of *j175* expression in seeds at different days after pollination revealed a trend characterized by an initial increase followed by a subsequent decrease, with the peak expression level observed at 10 DAP (Figure 3G).

### 2.10. Transcriptome Profiling of j175

To analyze the gene expression differences between *j175* and the WT, RNA-seq was performed on ear leaves and kernels. A large number of differential genes were detected (Figure 4A). A clustering heatmap analysis of the differential genes reveals that replicates from each sample are grouped closely, indicating similar expression patterns among replicates (Appendix A).

KEGG (Kyoto Encyclopedia of Genes and Genomes) pathway analysis indicated that differentially expressed genes were mainly concentrated in metabolic and photosynthetic pathways, including “sucrose and starch metabolism,” “carbon metabolism,” and “photosynthesis,” with downregulation being more prevalent than upregulation. The upregulated genes were primarily concentrated in pathways such as plant–pathogen interactions, plant hormone signal transduction, and the MAPK signaling pathway. The different leaves of J175 showed varying effects on plant metabolism, hormone signal transduction, and MAPK plant disease resistance signaling pathways (Figure 4B and Appendix A). Similarly to the leaf transcriptome, KEGG analysis of the kernel transcriptome showed that differentially expressed genes are in the pathway sets of “plant hormone signal transduction” and “plant pathway interaction.” However, the difference is that the kernel differential genes are largely enriched in “protein processing in endoplasmic reticulum” (Figure 4C).

The expression of plastid division-related genes in the green and white parts of *j175* leaves was inconsistent. Most plastid division-related genes, including *j175*, were downregulated in the green leaf parts and kernels of *j175*, while the upregulation of *FtsZ2-1* may partially complement the *j175* gene mutation. Additionally, the expression levels of *ARC3* and *GC1*, which are negative regulators of chloroplast division, were significantly increased in the mutant (Figure 4E). To further verify the reliability of the transcriptome data, we conducted qPCR validation of several genes, and the results were consistent with the transcriptome data (Appendix A). Many essential genes involved in kernel development were also significantly differentially expressed between the WT kernels and *j175* kernels. Among these, genes such as *smk9* (small kernel opaque endosperm2), *o2* (opaque endosperm2), *fl2* (floury endosperm2), *fl3*, *fl4*, and *bt2* (brittle endosperm) were downregulated in *j175* kernels. However, genes such as *betl3* (basal endosperm transfer layer3), *betl4*, *betl9*, *betl10*, and *bap2* (basal layer antifungal protein 2) were upregulated in *j175* kernels (Figure 4D). To explore the significant differences in chlorophyll content between the WT and *j175*, we analyzed the expression levels of key enzyme genes involved in the chlorophyll synthesis pathway. In Figure 3F, the expression of the *CHLH* gene, which encodes the H subunit of magnesium chelatase, was significantly downregulated in the white parts of J175 leaves. Together, CHLH, CHLI, and CHLD form a magnesium-chelating enzyme (MgCh) that catalyzes the transformation of protoporphyrin IX into magnesium protoporphyrin IX. This enzyme is crucial in chlorophyll synthesis, and the mutation in the *j175* gene also inhibits this process.

### 2.11. SNPs Associated with ZmFtsZ2-2 Can Increase Starch Content

This study utilized the association population of 280 maize inbred lines to analyze the candidate gene Zmj175 for grain-related traits, in order to investigate the relationship between natural variation in these genes and grain development, and to identify excellent natural alleles, providing valuable gene resources for high-yield molecular breeding of maize.

Candidate gene association analysis of related traits of endosperm starch content in grains showed that multiple natural-variation loci of j175 were significantly associated with grain length; the *p* values of SNP151150703, SNP151152523, SNP151152541, SNP151152883, and SNP151152954 in the coding region indicated extremely significant relationships, and they were in strong linkage disequilibrium (R^2^ > 0.9). These five SNP loci divided the maize inbred lines into two haplotypes: Hap1 and Hap2. The average starch content of the Hap1 haplotype in 147 inbred lines was 70.25%, and that of the Hap2 haplotype in 53 inbred lines was 67.69%. The starch content of Hap1 was significantly higher than that of Hap2 (*p* = 4.11 × 10^−3^). These four SNPs are all located in the coding region, but only the polymorphic amino acids of SNP151152883 are changed. There are two types of amino acid residues at this site: glutamine and histidine. Hsp1 is glycine and hsp2 is serine, indicating that the glycine at this site may be beneficial for the formation of longer grains, while hsp2 is an excellent haplotype (Figure 5).

## 3. Discussion

The leaf is the site where photosynthetic products are synthesized, determining the production of assimilates and acting as the “source” of yield formation [23]. As an important C4 plant, 90% of corn yield is derived from leaf photosynthesis [24,25]. Grain serves as the “reservoir” for materials stored in plants, with starch being the most important storage material in grains, accounting for approximately 70% of the total content [26]. We discovered that an SNP mutation in the exon of *j175* (Figure 3A) prevented chloroplast division and inhibited chlorophyll synthesis (Figure 1B,K–O). Thus, this reduced the photosynthetic efficiency in *j175* (Figure 1F–I). Simultaneously, this mutation limited the division of the amyloplasts (Figure 2P–S), altering the shape of starch granules and decreasing the total starch content (Figure 2M–O). Therefore, the *j175* mutation affects the “source” and “sink” of yield composition, resulting in a significant reduction in most agronomic traits of this material (Figure 2D–L). *j175-1* and *j175-2* are allelic mutants of *j175*; due to the higher position of the *j175-1* mutation site, more structural domain functions are lost, resulting in a more severe reduction in agronomic traits for *j175-2* (Appendix A). This result further proves that the *j175* mutation has a certain impact on the agronomic traits of maize.

Chloroplast division mutants are essential genetic materials for studying the underlying mechanisms of chloroplast division in plants [2,9]. Partial genes regulating chloroplast division have been identified, and their regulatory mechanisms have been elucidated through studies of Arabidopsis mutants deficient in chloroplast division [13,27,28,29]. This includes FtsZ1, ARC5, ARC6, and PDV1 (plastid division 1). In Arabidopsis, AtFtsZ directly or indirectly affects the localization and assembly of other proteins in the division complex, thereby inhibiting chloroplast division [8,12]. An increase in AtFtsZ1-1 protein levels inhibits chloroplast division, with higher AtFtsZ1-1 protein levels resulting in more severe phenotypes [16]. Arabidopsis FtsZ2-1 and FtsZ2-2 are functionally redundant [30]; we found, through evolutionary tree analysis, that AtFtsZ2-1 and AtFtsZ2-2 have the highest similarity of 83.55%, while ZmFtsZ2-2 and ZmFtsZ2-1 have only 63.11% similarity. ZmFtsZ2-1 has the highest sequence similarity with SbFtsZ2-2 in sorghum. This implies that maize ZmFtsZ2 has different functions compared to Arabidopsis AtFtsZ2 (Figure 3B and Appendix A). The *j175* mutant gene found in maize not only affects chloroplast division, as observed in other related genes, but also causes yellowing of *j175* seedlings, weak leaf growth, and the formation of white stripes on the leaves, affecting plant growth (Figure 1A–F and Appendix A). This suggests that the *j175* gene may serve different functions across various species. The high expression of j175 gene in green tissues such as the ears, ligules, and bracts further indicate its potential role in regulating plant growth and development (Figure 3F,G). In the mesophyll cells of *j175*, the chloroplasts were larger and fewer compared to the wild-type chloroplasts (Figure 1K–O), suggesting that mutations in the FtsZ 2-2 protein encoded by j175 cause damage to the Z-ring structure during maize chloroplast division, thereby hindering the division process. Furthermore, the RNA-seq and qPCR results demonstrated that most genes related to chloroplast division are inhibited to varying degrees in the *j175* green parts leaves (Figure 4E and Appendix A), supporting this hypothesis. Interference with chloroplast development may affect chlorophyll synthesis, and genes involved in chlorophyll biosynthesis (including *ZmHEMD, ZmCHLH, ZmCHLD, and ZmPORD*) are inhibited in the mutants (Figure 4F). As the production sites of salicylic and jasmonic acid, which are important mediators of plant immunity, chloroplasts also participate in PAMP (pathogen-associated molecular pattern)-induced defense gene expression, playing a vital role in plant immunity [31,32]. Therefore, in the GO and KEGG pathway analyses, several genes related to hormone signal transduction and plant-pathogen interactions were downregulated (Figure 4B,C and Appendix A). Additionally, the significant reduction in peroxidase activity observed in the enzyme activity assay could also indicate that the stress resistance of *j175* was weakened (Figure 1J). Thus, mutations in genes related to chloroplast division may reduce plant immunity.

Arabidopsis FtsZ2-1 and FtsZ2-2 are functionally redundant. Unlike the compound starch granules found in rice endosperm, maize endosperm contains simple starch granules, with only one starch granule per amyloplast [33,34]. Changes in the expression levels of amyloplast division proteins can alter starch granule synthesis [35]. Moreover, the membrane structure of amyloplasts is also affected by the levels of plastid division proteins [36]. In the *j175* mutant, amyloplast division was inhibited during growth and maturity, and amyloplasts in the mature grain endosperm existed in the form of oligomers, with a significant decrease in quantity (Figure 2P–S). An increase in StFtsZ1 protein levels in potatoes resulted in a significant reduction in the number and size of starch granules in tubers [37]. Similarly, in FtsZ-deficient Arabidopsis mutants, amyloplasts did not proliferate [38,39]. Meanwhile, the *j175* mutant in maize exhibited a change in amyloplast shape, implying that the regulation of the maize *j175* gene in amyloplasts differs from that of Arabidopsis and potato. Normal amyloplast division, such as during gravity sensing, is essential for plant growth and development [40,41]. Amyloplasts in Arabidopsis are essential for root gravity sensing and for repolarizing LAZY proteins through sedimentation [42,43,44,45]. Whether the *j175* gene in maize regulates gravity sensing in plants still requires further study. The formation of vitreous/opaque endosperm depends on the close interactions between proteosomes (storage gliadin) and amyloplasts [46]. Most opaque and floury genes in maize regulate zein synthesis by influencing the regulatory or structural genes of zein [46,47,48,49]. The upregulation of *o2, fl2*, *fl4*, *de30*, and other genes in the kernels of the *j175* mutant may serve as compensation for the deletion of the *j175* gene, allowing stable synthesis of gliadin (Figure 4D). Through gene association analysis, it was found that there were several SNPs on zmftsz2-2 that were significantly associated with starch content, which may also suggest that this gene can participate in the regulation of starch (Figure 5). However, this specific interaction requires further investigation. The expression of all basal endosperm transfer layer genes and basal antifungal protein genes in maize was downregulated in the j175-kernel (Figure 4D). This finding implies that the *j175* gene plays a critical role in regulating the development and immunity of the maize endosperm transfer layer, which may contribute to the smaller kernel size observed in j175 (Figure 2D–L).

## 4. Conclusions

The white-stripe mutation of maize is caused by a single-nucleotide substitution in the exon of the filamentous temperature-sensitive Z protein gene, *j175*. This mutation affects the division of chloroplasts and amyloplasts, thereby affecting agronomic traits.

## 5. Materials and Methods

### 5.1. Cultivation of Materials and Investigation of Agronomic Traits

The leaf color-related mutant, isolated from the EMS mutant library with an RP125 background [50], was temporarily named *j175* after undergoing continuous self-selection to stabilize its traits. The material was cultivated and managed in the provinces of Wenjiang, Chengdu, Sanya, and Hainan. We observed the phenotypes of *j175* plants at different stages. During the maturity period, we measured agronomic traits for mutant and wild-type plants, including plant height, ear height, weight, leaf length, leaf width, leaf thickness, ear length, ear thickness, ear weight, ear shaft thickness, ear shaft weight, grain length, grain width, grain thickness, and hundred-grain weight. The data on these agronomic traits were analyzed using Prism 10.2 software.

### 5.2. Chlorophyll Content Measurement

To determine the chlorophyll content, we collected 0.1 g of green leaf parts and white leaf parts of j175, as well as wild-type leaves, at the heading stage, and created sets of 6 replicates. Chlorophyll was measured using a UV-visible spectrophotometer (UV-5200, China). We used the following equations to calculate the chlorophyll content:Chl *a* (mg g^−1^) = (12.7D_663_ − 2.69D_645_) × (100/0.1 × 1000)Chl *b* (mg g^−1^) = (22.9D645 − 4.68D663) × (100/0.1 × 1000)Total Chlorophyll (mg g^−1^) = Chl *a* (mg g^−1^) + Chl *b* (mg g^−1^)

### 5.3. Determination of Photosynthetic Rate

A portable photosynthesis analyzer (Li-COR, Li-4800, Lincoln, NE, USA) was used to measure the photosynthetic rates of the white parts and green parts of *j175* leaves and that of wild-type leaves during the jointing stage. Measurements included the transpiration rate, net photosynthetic rate, stomatal conductance, and intercellular CO_2_ concentration. Six biological replicates were established for each measurement.

### 5.4. Enzyme Activity Assessment

The white parts and green parts of j175 leaves, as well as wild-type ear leaves, were collected during the jointing stage, and sets of 6 replicates were created. The following kits were utilized to measure the enzyme activity in the leaves: a plant-soluble sugar content detection kit (BC0030, Solaibao, Beijing, China), a malondialdehyde (MDA) content detection kit (BC0020, Solaibao, Beijing, China), a catalase (CAT) activity detection kit (BC0200, Solaibao, Beijing, China), and a peroxidase (POD) activity detection kit (BC0090, Solaibao, Beijing, China).

### 5.5. Chloroplast Morphology Observation

The white parts and green parts of j175 leaves, as well as wt ear leaves, were collected during the jointing stage and subjected to a series of processes, including fixation, dehydration, infiltration, embedding, ultrathin sectioning, and staining. Finally, the fixed leaves were examined under a transmission electron microscope (JEM-1400FLASH, Tokyo, Japan) at various magnifications (200×, 500×, and 1000×).

### 5.6. Observation of Amyloplast and Starch Granule Morphology

Mutant and wild-type kernels were collected at 12 DAP (Days After Pollination) and 20 DAP. The endosperms were then placed in a culture dish, and a separation solution (50 mmol/L Tris, 0.8 mol/L sucrose, 1 mmol/L disodium EDTA, and 1 mmol/L KCl, at pH 7.5, with 0.1% bovine serum albumin and 1% mercaptoethanol added when used) was added. Furthermore, the endosperm was cut into small particles using a blade and filtered. A pipette was used to absorb the homogenate surrounding the endosperm, and this process was repeated twice before centrifuging the filtrate. The supernatant was discarded, and 5 mL of the separation solution was slowly added while gently flipping the centrifuge tube to suspend the precipitate; this process was repeated once or twice. Following centrifugation, the precipitate obtained was amyloplasts. Subsequently, 2 mL of the separation solution was added dropwise, followed by a drop of 0.5% I2-KI (Liquor Iodi et Kalii Iodidi) solution, and the sample was observed under a fluorescence microscope [51,52,53].

Mature kernels of J175 and wild-type plants were selected from the middle of the ear. Each kernel was cut in half from top to bottom with a blade to reveal the complete embryo and endosperm, and then fixed onto the operating plate with the cutting surface facing upwards. Gold powder was sputtered onto the cut surface of the kernel to enhance its conductivity, and then the kernel was exposed to a vacuum for 30 min. Additionally, the sections were observed at different magnifications (500×, 1000× and 2000×) using a scanning electron microscope (Sigma 500, Zeiss, Jena, Germany).

### 5.7. Map-Based Cloning and Allelic Test

The inbred line B73 is one of the representative varieties of the maize genome; we hybridized the *j175* mutant with B73 to generate an F2 segregating population. Genetic analysis was performed by calculating the segregation rate between plants for the mutant phenotype and wild-type plants in the F2 generation, with a chi-squared test used to confirm the fit. DNA from isolated single plants in the F2 generation was used for positional cloning [54]. For the initial mapping of the *J175* gene, an isolated F2 population was utilized to select 50 leaves with significant phenotypes and 50 leaves without phenotypes, creating mixed dominant and recessive pools for BSA-seq analysis. The population was subsequently expanded, and new polymorphic molecular markers were screened to enable more precise mapping of linkage genes related to the mutant phenotype by PCR amplification of RP125, j175, F1, and the aforementioned mixed-pool DNA. Molecular markers were specifically designed to localize accurately based on polymorphisms between the reference genomes of B73 and RP125 (Appendix A). To confirm the candidate gene obtained by map-based cloning, we performed an allelic test. An EMS-mutagenized allelic mutant (EMS5-099e01, *j175-1*; EMS4-0e87c8, j175-2) of *j175* with a B73 inbred line background was obtained from maizeEMSDB [55], and the allelic test was conducted using *j175*, *175-1*, and *j175-2*.

### 5.8. Protein Sequence Analysis

Protein sequences were aligned using the ClustalW model of MEGA11 [56] (Appendix A). The gene structure diagram was generated using the online website GSDS2.0 [57]. The input protein sequence of the structural domain was conserved for Batch CD Search in NCBI, and the output file was processed for visualization analysis using the TBools-II tool [58]. The protein motif structures were predicted using MEME [59].

### 5.9. Subcellular Localization

To determine the subcellular localization of the J175 protein, its full-length coding sequence (excluding stop codons) was amplified and cloned into the pCAMBIA2300 subcellular localization expression vector, which includes eGFP tags to construct the C-terminal fusion protein J175-eGFP. The 35S: J175 eGFP and 35S: eGFP plasmids were then introduced into maize protoplasts via a polyethylene glycol (PEG)-mediated transformation method. eGFP fluorescence was subsequently detected using a laser confocal scanning microscope (LSM880, ZEISS, Jena, Germany).

Using NCBI to perform quantitative primer design, we first screened the designed primers for specificity, and used actin as an internal reference to analyze the tissue expression specificity of materials at different time points (Appendix A).

### 5.10. RNA-Seq Analysis

Samples were collected from the middle of the white, green, and wild-type spike leaves of the jointing mutant. Mutant and wild-type kernels at 10 days after pollination (DAP) were selected, and the kernel coat was removed. Three independent replicates were obtained for each sample. Total RNA was extracted with the Trizol reagent (15596026, Solaibao, Beijing, China) and an Ultrapure RNA Kit (R2230, Solaibao, Beijing, China). Library construction and sequencing were completed by Anno by Grand Omics (Wuhan, China). For the gene expression results, the differentially expressed genes among samples were screened out, and GO function significance enrichment analysis and KEGG pathway significance enrichment analysis were carried out.

## Figures and Tables

**Figure 1 plants-14-02198-f001:**
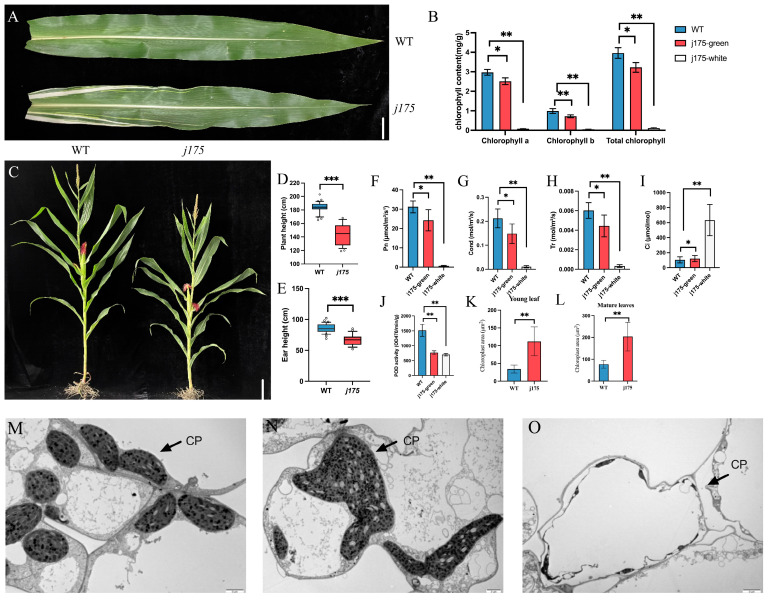
Leaf phenotypes of *j175*. (**A**) Ear leaf phenotypes of WT and *j175*. Scale bar: 4 cm. (**B**) From left to right are the chlorophyll a content, chlorophyll b content, and total chlorophyll content; the three samples are a WT ear leaf, the green part of a *j175* ear leaf, and the white part of a *j175* ear leaf (*n* = 6). (**C**) Mature plant phenotypes of WT and *j175*. Scale bar, 10 cm. (**D**,**E**) The plant height (**D**) and ear height (**E**) of mature WT and *j175* plants (*n* = 10). (**F**–**I**) Photosynthetic rate-related indicators: Pn (net photosynthetic rate) (**F**), Cond (stomatal conductance) (**G**), Tr (transpiration rate) (**H**), and Ci (intercellular CO_2_ concentration) (**I**); the three samples are a WT ear leaf, the green part of a *j175* ear leaf, and the white part of a *j175* ear leaf (*n* = 6). (**J**) Enzyme activity assay: POD (peroxidase); the three samples are a WT ear leaf, the green part of a *j175* ear leaf, and the white part of a *j175* ear leaf (*n* = 6). (**K**,**L**) Statistical analysis of chloroplast surface area in leaf mesophyll cells of WT and j175 plants during the seedling (**K**) and mature stages (**L**) (*n* = 20). (**M**–**O**) Observation of chloroplasts in the mesophyll cells of ear-position leaves. From left to right: the WT (**M**), the green part of *j175* leaves (**N**), and the white part of *j175* leaves (**O**). Scale bar: 2 μm. Data are presented as the mean ± SD and were statistically analyzed using Student’s *t*-test. * (*p* < 0.05) denotes a significant difference between the WT and *j175*, ** (*p* < 0.01) indicates a highly significant difference, and *** (*p* < 0.001) indicates an extremely significant difference.

**Figure 2 plants-14-02198-f002:**
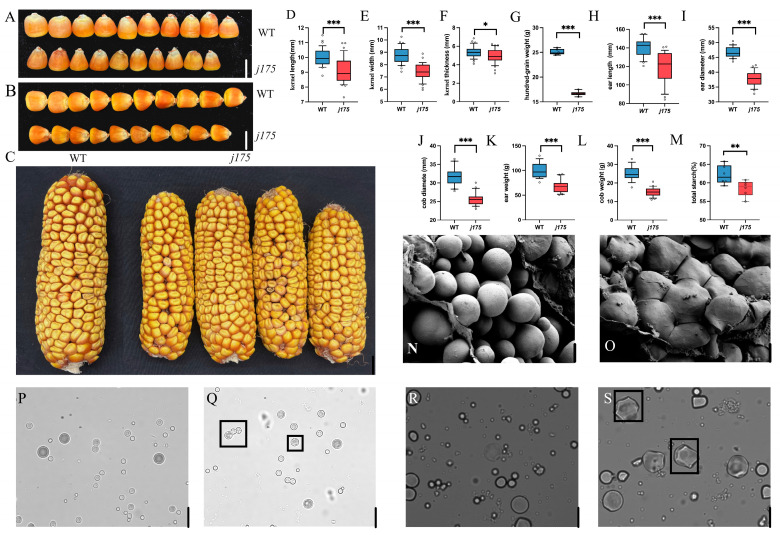
Kernel phenotypes of *j175*. (**A**,**B**) The kernel width (**A**) and kernel length (**B**) of the WT and j175. Scale bar, 1 cm. (**C**) Ear phenotypes of the WT and *j175*. Scale bar: 2 cm. (**D**–**G**) Kernel agronomic traits: the kernel length (**D**), kernel width (**E**), kernel thickness (**F**), and hundred-kernel weight (**G**) of the WT and *j175* (*n* = 20). Ear agronomic traits: the ear length (**H**), ear diameter (**I**), cob diameter (**J**), ear weight (**K**), and cob weight (**L**) of the WT and *j175* (*n* = 20). (**M**) The kernel endosperm starch content of the WT and *j175* (*n* = 6). (**N**,**O**) Differences in the shape of starch granules between the WT (**N**) and *j175* (**O**). (**P**,**Q**) Endosperm amyloplasts of the WT (**R**) and *j175* (**S**) at 12 DAP. (**R**,**S**) Endosperm amyloplasts of the WT and *j175* at 20 DAP. (**N**–**S**) Scale bar: 5 μm. black square: Amyloplasts with different shapes. Data are presented as the mean ± SD and were statistically analyzed using Student’s *t*-test. * (*p* < 0.05) denotes a significant difference between the WT and *j175*, ** (*p* < 0.01) indicates a highly significant difference, and *** (*p* < 0.001) indicates an extremely significant difference.

**Figure 3 plants-14-02198-f003:**
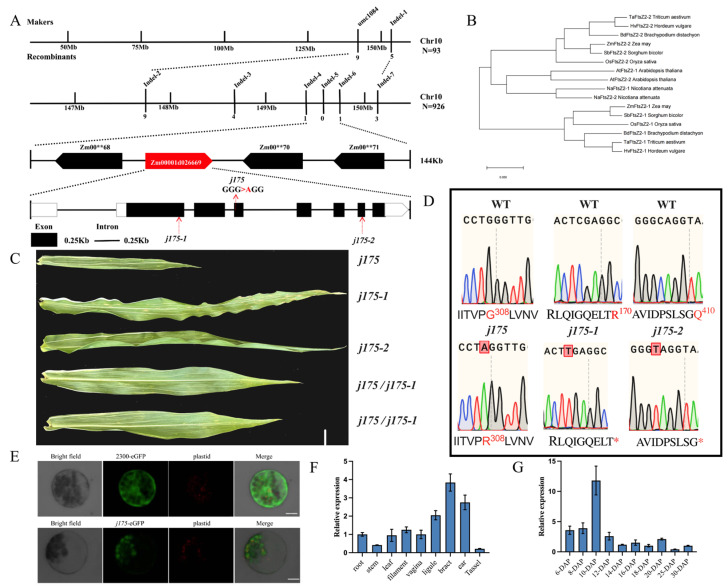
*Zm00001d026669* is *j175*. (**A**) Upper part: positional cloning of *j175*. The four boxes represent four genes, and the candidate gene is in red. Lower part: the gene model of *Zm00001d026669* and mutation site of *j175*. The black boxes represent exons, the black lines represent introns, and the white boxes represent untranslated regions. The red arrows indicate *j175*, *j175-1*, and *j175-2* mutation sites. (**B**) Evolutionary tree analysis of FtsZ2-1 and FtsZ2-2. The species involved in the analysis were Zea mays, Arabidopsis thaliana, Oryza sativa, Nicotiana tabacum, Sorghum bicolor, Triticum aestivum, Brachypodium distachyon, and Hordeum vulgare. (**C**) Allelic testing, from top to bottom, of *j175*, *j175-1*, *j175-2*, *j175/j175-1*, and *j175/j175-2*. Scale bar: 5 cm. (**D**) Sequence peaks of the WT, *j175*, *j175-1* and *j175-2* at mutation sites. Red letters, mutant amino acids (up) and bases (below). (**E**) Subcellular localization. Scale bar: 10 μm. (**F**) J175 group-forming expression. (**G**) Kernel expression of j175 at different DAP.

**Figure 4 plants-14-02198-f004:**
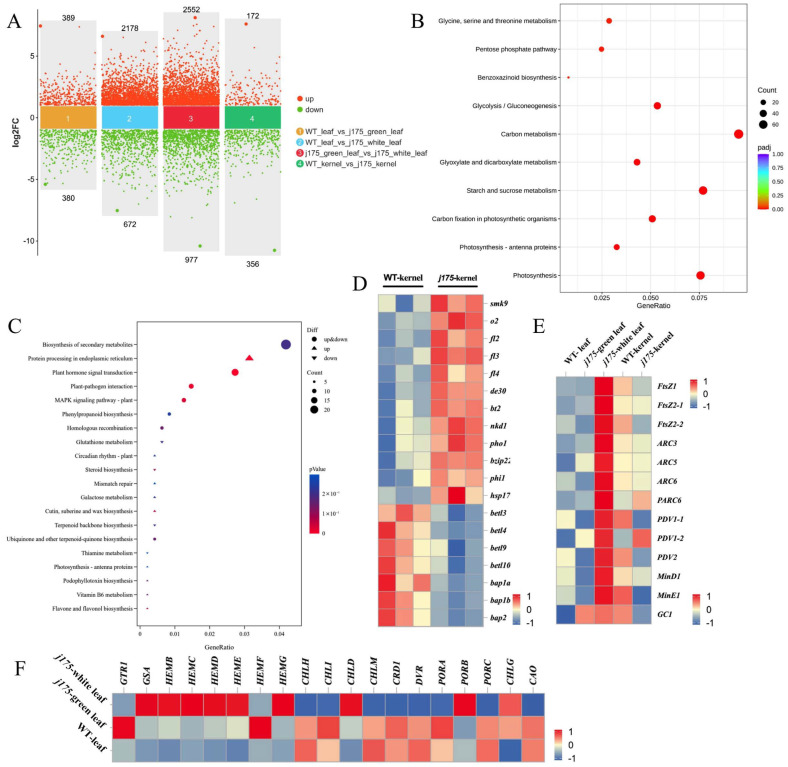
Global transcriptome analysis based on RNA-seq data from leaves and grains of the WT and *j175*. (**A**) A scatter diagram of different genes in multiple groups: WT leaf vs. *j175* green partial leaf, WT leaf vs. *j175* white partial leaf, *j175* green partial leaf vs. *j175* white partial leaf, and WT kernel vs. *j175* kernel. (**B**) KEGG enrichment analysis of WT leaf vs. j175 green partial leaf (**B**) and WT kernel vs. *j175* kernel (**C**). (**D**) Expression levels of genes involved in kernel development. (**E**) Expression of genes related to chloroplast division. (**F**) Expression of genes related to chlorophyll synthesis.

**Figure 5 plants-14-02198-f005:**
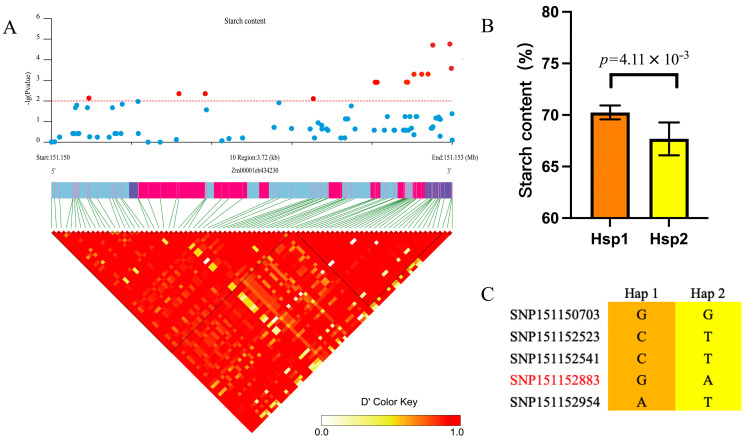
Haplotypes of ZmFtsZ2-2 among natural inbred lines of maize. (**A**) Association analysis of natural variations in ZmFtsZ2-2. The upper region indicates the gene’s associated SNP sites. The blue dots below the log value are non-significant SNPs, and the red dots above the log value are significant SNPs. The lower part is the pattern of pairwise linkage disequilibrium of the variations. (**B**) The starch content of the different haplotypes of ZmFtsZ2-2. *n* = 147 for Hap 1; *n* = 56 for Hap 2. (**C**) The haplotype information of ZmFtsZ2-2; black SNPs are the same mutation, and red SNPs are missense mutations. Data are presented as the mean ± SD and were statistically analyzed using Student’s *t*-test.

## Data Availability

The data presented in this study is available on request from the corresponding author.

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
