# Peer review of "Filamentous Temperature-Sensitive Z Protein J175 Regulates Maize Chloroplasts’ and Amyloplasts’ Division and Development"

_plants, 2025, doi:10.3390/plants14142198_

Round 1

Reviewer 1 Report

Comments and Suggestions for Authors

The work is devoted to the problem of regulation of chloroplast and amyloplast division in maize plants. The issue is very important and not new. In a number of plant species, genes have been identified that are responsible for affect chloroplast division. These studies were conducted on a very important agricultural crop - maize. In addition, maize belongs to the C4-type of photosynthesis plants. The second major advantage of this work is that the authors gave an exceptionally comprehensive description of the influence of the J175 gene mutation not only on plastid division, but also on plant development, yield, activity of many enzymes, polysaccharides, etc. This allowed the authors to conclude that the gene under study plays an important role in various aspects of maize growth and development.

The work is good and I will be glad if, after editing, it is published in the Plants journal.

There are comments. I provide my questions without dividing them by importance. The lines to which the questions relate are indicated on the left.

Section 2.10 The authors describe in some detail such a simple method as determining the chlorophyll content and there is no information at all about a very complex methodological approach such as RNA-Seq Analysis. It is clear that the authors themselves did not create or analyze the transcriptomes, but at least some generally accepted characteristics of the transcriptomes should be given. The company probably does a good job of this, but this does not mean that readers should take the authors' word for it.

171 There is no information on obtaining corn protoplasts

208-209 The authors should talk about the white parts of the mutant leaves, not the white leaves of the mutants. The caption to Figure 1 is written correctly, but this error is repeated many times in the text.

242-244 Why do the authors think that the absence of difference between WT and J175 in SOD dismutase activity and proline content indicates a decrease in the adaptive potential of mutant plants? Most likely, it indicates that the supposed resistance of plants to abiotic stressors has not changed.

350 Overall, 35,375 genes were detected. And what is this number of genes? This section pays much attention to the number of genes activated and/or suppressed in different plants. This is not very interesting and not very useful information. It can be shortened.

334 Subcellular Localization and Constitutive Expression of j1754. In this section, constitutive expression is not confirmed.

The authors suggest, and not without reason, that a mutation in 1 nucleotide with a G to A substitution led to such a serious violation. To prove this conclusion more clearly, it would be a good idea to transform mutant plants with the non-mutated J175 gene.

Some phrases or fragments are not written very clearly (at least for me)

14  J175 … expressed across various tissues. It is not the protein that is expressed, but the gene.

106 CTChlorophyll What is this?

128 The seeds of wild-type and mutant plants were pollinated for 12 and 20 days, respectively. I do not understand this phrase.

138 Mature seeds of J175 and wild-type plants were selected from the same location. It would be better to write more clearly. What location are we talking about?

391-396 are not written very clearly.

Figure 2. The letter N in Figure 2 is completely invisible. It is better to transfer it to a lighter part of the Fig. 2.

Author Response

Comments 1: Section 2.10 The authors describe in some detail such a simple method as determining the chlorophyll content and there is no information at all about a very complex methodological approach such as RNA-Seq Analysis. It is clear that the authors themselves did not create or analyze the transcriptomes, but at least some generally accepted characteristics of the transcriptomes should be given. The company probably does a good job of this, but this does not mean that readers should take the authors' word for it.

Response 1: Thank you for your advice. The detailed description of 2.2 chlorophyll extraction simple method is indeed redundant and has been simplified. The analysis of 2.10 transcriptome, which is too simple, has been carefully modified.

Comments 2: 171 There is no information on obtaining corn protoplasts

Response 2: Thank you for your advice. “Pro toplasts were obtained from the leaves of etiolated seedlings after 2 weeks of germination. Photos were captured by laser scanning confocal microscopy (LMS880, Zeiss)” has been added in this section.

Comments 3: 208-209 The authors should talk about the white parts of the mutant leaves, not the white leaves of the mutants. The caption to Figure 1 is written correctly, but this error is repeated many times in the text.

Response 3: Thank you for your advice. “white leaves of the mutants and green leaves of the mutants” in the text have been changed to “white parts of the mutant leaves and green parts of the mutant leaves”.

Comments 4: 242-244 Why do the authors think that the absence of difference between WT and J175 in SOD dismutase activity and proline content indicates a decrease in the adaptive potential of mutant plants? Most likely, it indicates that the supposed resistance of plants to abiotic stressors has not changed.

Response 4: Thank you for your advice. There is no change in superoxide dismutase and proline in the leaves of the two plants, which is meaningless to draw a conclusion. We will delete this part. However, differences in peroxidase, malondialdehyde, catalase and soluble sugars do affect the stress resistance of plants.

Comments 5: 350 Overall, 35,375 genes were detected. And what is this number of genes? This section pays much attention to the number of genes activated and/or suppressed in different plants. This is not very interesting and not very useful information. It can be shortened.

Response 5: Thank you for your advice. This section is indeed redundant and has been carefully deleted.

Comments 6: 334 Subcellular Localization and Constitutive Expression of j1754. In this section, constitutive expression is not confirmed.

Response 6: Agree. The description that the constitutive expression of j175 is confirmed has been added in this section. “ J175 were constitutive expression and expressed across various tissues, with the highest expression levels in the bracts and female tassels and the lowest in the male tassels (Fig. 3F)”.

Comments 7: The authors suggest, and not without reason, that a mutation in 1 nucleotide with a G to A substitution led to such a serious violation. To prove this conclusion more clearly, it would be a good idea to transform mutant plants with the non-mutated J175 gene.

Response 7: Thank you for your suggestion. We confirmed that j175 is a candidate gene for this phenotype through allelic validation in Figure 3C, and this statement was also supported by Supplementary Figure 3. Using the non mutated J175 gene to transform mutant plants is indeed a good idea, which can once again confirm this conclusion from another perspective. We hope you understand our concern that we were unable to complete the above experiment during the quality review cycle of this journal.

Comments 8: Some phrases or fragments are not written very clearly (at least for me)

Response 8: This is a good suggestion. Thank you for your suggestion. I will revise the article again.

Comments 9: 14  J175 … expressed across various tissues. It is not the protein that is expressed, but the gene.

Response 9: Thank you for your advice. The two cannot be abbreviated. It has been revised to ”J175 protein is localized in plastids ,and its gene expressed across various tissues”.

Comments 10: 106 CTChlorophyll What is this?

Response 10: Thank you for your advice. “Ctchlorophyll” is indeed prone to ambiguity. I have replaced it with “Total Chlorophyll”.

Comments 11: 128 The seeds of wild-type and mutant plants were pollinated for 12 and 20 days, respectively. I do not understand this phrase.

Response 11: Thank you for your advice. Modified to "Kernels of mutant and wild-type 12 DAP (Days After Pollination) and 20 DAP were collected."

Comments 12: 138 Mature seeds of J175 and wild-type plants were selected from the same location. It would be better to write more clearly. What location are we talking about?

Response 12: Thank you for your advice. I have changed “the same location” to “the middle of the ear”.

Comments 13: 391-396 are not written very clearly.

Response 13: Agree. The enrichment analysis of the transcriptome has been described too much, and this pair has been streamlined.

Comments 14: Figure 2. The letter N in Figure 2 is completely invisible. It is better to transfer it to a lighter part of the Fig. 2.

Response 14: Thank you for your advice. I changed the position of the letters N and O.

Reviewer 2 Report

Comments and Suggestions for Authors

In this paper, the authors examine the J175 (FtsZ2-2) gene, cloned from an ethyl methanesulphonate (EMS) mutant involved in chloroplast and amyloplast division in maize, through map-based cloning. They found that J175 encodes a cell division protein, FtsZ (filamentous temperature-sensitive z). The authors argued that J175 is localized in plastids and expressed across various tissues. From the seedling stage, the leaves of the j175 mutant exhibited white stripes, while the division of chloroplasts was inhibited, leading to a significant increase in volume and a reduction in their number. The photosynthetic rate showed a significant decrease in the photosynthetic efficiency of j175. In addition, the division of amyloplasts in j175 grains at different stages was impeded, resulting in irregular polygonal starch granules. RNA-seq analyses of leaves and kernels showed that multiple genes affecting plastid division, such as FtsZ1, ARC3, ARC6, PDV1-1, PDV2, and MinE1, were significantly downregulated.

The introduction is well structured, but there are some minor edits.

P2 Line 74 - Before discussing the aims of this study on the J175 genes, it is important to briefly explain why J175 genes are relevant, or if there are other studies about them. If this is the first report, it is also important to mention that to establish the relevance of this research.

In the Materials and Methods section, it is important to elaborate.

P2 Line 86 - Where was the EMS mutant library sourced from? Provide elaboration or a reference.

P3 Line 96 - Was green and white tissue taken only from mutants, and green tissue only from wild type? Please clarify (like: The three samples are WT ear leaf, j175 green part ear leaf, and j175 white part ear).

P3 Line 96 -“The leaves were washed dried” a “and” is missing. Change to “The leaves were washed and dried”.

P3 Line 102 - Indicate the number of biological and technical replicates analyzed for chlorophyll content measurement.

P3 Line 114 - Specify the number of biological and technical replicates analyzed for the enzyme activity assays.

P3 Line 128 - Include details of the reagent used to prepare the separation solution or provide a reference.

P4 Line 139 - Elaborate on the sample preparation process. Include specifics about the microtome and details on the sample thickness.

P4 Line 144 - Indicate in one line why B73 was used for crossing with the j175 mutant.

P4 Line 147 - Specify the methodology used to extract genomic DNA or include a reference.

P4 Line 160 -Specify which protein sequences were used to align and generate the structural domain.

P4 Line 178 - Before mentioning that the samples were sequenced, specify which method of RNA extraction was used or provide a reference. Once the RNA-Seq was done, clarify what type of analysis was used for differential expression. Provide a brief explanation of how the gene ontology was constructed and include the statistics used.

In the Results section

P5 Line 192 - In “green part ear leaf, j175 white part ear leaf” an “and” is missing, so it should say green part ear leaf, and j175 white part ear leaf (n=5).

P5 Line 195 - Tr (transpiration rate) (G) refers to H, and Cond (stomatal conductance) (H) refers to G.

P5 Line 197 - The three samples are WT ear leaf, j175 green part ear leaf, “and” j175 white part ear leaf (n=5). This is repeated exactly in line 196. Add the “and”.

P5 Figure 1. Indicate the localization of the chloroplasts in photos N and O with arrows.

P6 Line 217 - Indicate the significance value (e.g., P > 0.001).

P7 Figure 2. Add the scale bar to all photos from N to S.

P7 Line 258, 259, and 260. Scale bar 5μm, appears three times; keep only one.

P8 Line 291 - PCR conditions are missing from the Materials and Methods section. Please include this part.

P9 Line 327 - A space is needed after “site.B, Evolution”. It should be “site. B, Evolution”

P9 Line 328 – Cursive letters required for all scientific plant names.

P11 Line 396 - qPCR conditions for RNA-seq validation are missing from the Materials and Methods section. Please include this information and provide a list of the primers used.

P12 Line 422 – “The average starch content of Hap1 haplotype in 147 inbred lines was 70.25%, and that of Hap1 haplotype in 53 inbred lines was 67.69%. The starch content of Hap1 was significantly higher than that of Hap1”. It should be said that the average starch content of the Hap1 haplotype in 147 inbred lines was 70.25%, and that of the Hap2 haplotype in 53 inbred lines was 67.69%. The starch content of Hap1 was significantly higher than that of Hap2.

P12 Line 431 – In “lines.A”, a space is needed. It should say “lines. A,”

P12 Line 435 – “Histogram reveals…” Figure 5B is not a histogram. It should just say Starch content of the…

P13 Line 491 – There are unnecessary uppercase letters.

P17 Line 659 - In E-F, Mature…   “j175 mesophyll cells” is missing. “Are Functionally Redundant, Unlike”

The detected issues have been explained.

The authors should include the missing information to clarify the procedures used during the research and add the necessary details for this article to be published.

Author Response

The introduction is well structured, but there are some minor edits.

Comments 1: P2 Line 74 - Before discussing the aims of this study on the J175 genes, it is important to briefly explain why J175 genes are relevant, or if there are other studies about them. If this is the first report, it is also important to mention that to establish the relevance of this research.

Response 1: Thank you for your suggestion.  I have made relevant modifications, highlighting the first identification of this family of genes in maize.

In the Materials and Methods section, it is important to elaborate.

Comments 2: P2 Line 86 - Where was the EMS mutant library sourced from? Provide elaboration or a reference.

Response 2: Thank you for your suggestion. I have provided the references and relevant websites for the mutant library.

Comments 3: P3 Line 96 - Was green and white tissue taken only from mutants, and green tissue only from wild type? Please clarify (like: The three samples are WT ear leaf, j175 green part ear leaf, and j175 white part ear).

Response 3: Thank you for your advice. “white leaves of the mutants and green leaves of the mutants” in the text have been changed to “white parts of the mutant leaves and green parts of the mutant leaves”. And the entire text has been revised accordingly.

Comments 4: P3 Line 96 -“The leaves were washed dried” a “and” is missing. Change to “The leaves were washed and dried”.

Response 4: Thank you for your suggestion. I have made modifications here.

Comments 5:P3 Line 102 - Indicate the number of biological and technical replicates analyzed for chlorophyll content measurement.

Response 5: Thank you for your suggestion. I have added the number of technical and biological replicates.

Comments 6: P3 Line 114 - Specify the number of biological and technical replicates analyzed for the enzyme activity assays.

Response 6: Thank you for your suggestion. I have added the number of technical and biological replicates.

Comments 7: P3 Line 128 - Include details of the reagent used to prepare the separation solution or provide a reference.

Response 7: Thank you for your suggestion. I have added the detailed information of the separation solution.”[25]”

Comments 8: P4 Line 139 - Elaborate on the sample preparation process. Include specifics about the microtome and details on the sample thickness.

Response 8: Thank you for your suggestion. I have provided a more detailed description of the preparation of this part of the sample.

Comments 9: P4 Line 144 - Indicate in one line why B73 was used for crossing with the j175 mutant.

Response 9: Thank you for your suggestion. I have briefly explained the purpose of using B73 in this section.”The inbred line B73 is one of the representative varieties of maize genome”.

Comments 10: P4 Line 147 - Specify the methodology used to extract genomic DNA or include a reference.

Response 10: Thank you for your suggestion. I have described the DNA extraction method by adding references.”[29]”

Comments 11: P4 Line 160 -Specify which protein sequences were used to align and generate the structural domain.

Response 11: Thank you for your suggestion. I added protein sequence information for comparison in Supplementary 2.

Comments 12: P4 Line 178 - Before mentioning that the samples were sequenced, specify which method of RNA extraction was used or provide a reference. Once the RNA-Seq was done, clarify what type of analysis was used for differential expression. Provide a brief explanation of how the gene ontology was constructed and include the statistics used.

Response 12: Thank you for your suggestion. I have made detailed modifications to the transcriptome sequencing method.

In the Results section

Comments 13: P5 Line 192 - In “green part ear leaf, j175 white part ear leaf” an “and” is missing, so it should say green part ear leaf, and j175 white part ear leaf (n=5).

Response 13: Thank you for your suggestion. I have already changed it to “WT ear leaf, j175 green part ear leaf, and j175 white part ear leaf”.

Comments 14: P5 Line 195 - Tr (transpiration rate) (G) refers to H, and Cond (stomatal conductance) (H) refers to G.

Response 14: Thank you very much for your suggestion. I have corrected the error in that area.

Comments 15: P5 Line 197 - The three samples are WT ear leaf, j175 green part ear leaf, “and” j175 white part ear leaf (n=5). This is repeated exactly in line 196. Add the “and”.

Response 15: Thank you for your suggestion. I have already changed it to “WT ear leaf, j175 green part ear leaf, and j175 white part ear leaf”

Comments 16: P5 Figure 1. Indicate the localization of the chloroplasts in photos N and O with arrows.

Response 16: Thank you for your suggestion. I have marked the positions of chloroplasts for N and O in Figure 1.

Comments 17: P6 Line 217 - Indicate the significance value (e.g., P > 0.001).

Response 17: Thank you for your suggestion. I have added a description of * * (p<0.001).

Comments 18: P7 Figure 2. Add the scale bar to all photos from N to S.

Response 18: Thank you for your suggestion. I have adjusted the thickness of the proportion in the picture for better observation.

Comments 19: P7 Line 258, 259, and 260. Scale bar 5μm, appears three times; keep only one.

Response 19: Thank you for your suggestion. I have made the deletion and only retained a description on a scale.”N-S, Scale bar, 5 μm.”

Comments 20: P8 Line 291 - PCR conditions are missing from the Materials and Methods section. Please include this part.

Response 20: Thank you for your suggestion. I have added a section on PCR amplification in the Materials and Methods section.”The population was subsequently expanded, and new polymorphic molecular markers were screened to enable more precise mapping of linkage genes related to the mutant phenotype by PCR amplification of RP125, j175, F1, and the aforementioned mixed pool DNA.”

Comments 21: P9 Line 327 - A space is needed after “site.B, Evolution”. It should be “site. B, Evolution”

Response 21: Thank you very much for your suggestion. I have corrected the error in that area.

Comments 22: P9 Line 328 – Cursive letters required for all scientific plant names.

 Response 22: Thank you for your suggestion. I have made modifications to all the scientific plant names in Figure 3.

Comments 23: P11 Line 396 - qPCR conditions for RNA-seq validation are missing from the Materials and Methods section. Please include this information and provide a list of the primers used.

Response 23:  Thank you for your suggestion. I have added relevant information on qPCR in the Materials and Methods section and included the primer sequences in the attached table (Supplementary 2).

Comments 24: P12 Line 422 – “The average starch content of Hap1 haplotype in 147 inbred lines was 70.25%, and that of Hap1 haplotype in 53 inbred lines was 67.69%. The starch content of Hap1 was significantly higher than that of Hap1”. It should be said that the average starch content of the Hap1 haplotype in 147 inbred lines was 70.25%, and that of the Hap2 haplotype in 53 inbred lines was 67.69%. The starch content of Hap1 was significantly higher than that of Hap2.

 Response 24: Thank you for your suggestion. I have readjusted the word order of this paragraph as suggested.

Comments 25: P12 Line 431 – In “lines.A”, a space is needed. It should say “lines. A,”

Response 25: Thank you for your suggestion. I have corrected this error by adding a space.

Comments 26: P12 Line 435 – “Histogram reveals…” Figure 5B is not a histogram. It should just say Starch content of the…

Response 26: Thank you for your suggestion. I have modified the description of Figure 5B to “The starch content of the different haplotypes of ZmFtsZ2-2”.

Comments 27: P13 Line 491 – There are unnecessary uppercase letters.“Are Functionally Redundant, Unlike”

Response 27: Thank you for your suggestion. I have changed “Arabidopsis FtsZ2-1 and FtsZ2-2 Are Functionally Redundant, Unlike the compound starch granules found in the rice endosperm,” to “Arabidopsis FtsZ2-1 and FtsZ2-2 are functionally redundant, unlike the compound starch granules found in the rice endosperm”, and modified non essential capital letters.

Comments 28: P17 Line 659 - In E-F, Mature…   “j175 mesophyll cells” is missing.

Response 28: Thank you for your suggestion. I have added the missing content.

The detected issues have been explained.

Comments 29: The authors should include the missing information to clarify the procedures used during the research and add the necessary details for this article to be published.

Response 29: I have made corrections to all the problems mentioned in the articles above. Thank you again for your meticulous and patient suggestions.

Reviewer 3 Report

Comments and Suggestions for Authors

The paper is interesting and it investigates the location and function of a gene involved into  chloroplasts functionality. The work is solid and the results are well grounded. Minor concerns are related to the last part of RNA-seq analysis and functional mutations in j175 gene. 

Mutations j175-1 and j175-2 are loss of function mutations, Although the phenotype on figure 3 is strongly different from the phenotype of j175. The authors should explain this difference in phenotype and better explain why they relate the changes caused by j175-1 and j175-2  similar to the changes caused by j175. 

Transcriptome Profiling of j175: I think the entire section is less informative because the connection between the gene and expression of other genes, and in your case it is hundreds of genes, is unclear. How it is connected to chloroplasts division? Functional classification group support this? Figures 4 B,C are  less informative because of the low number of genes. 

Reduce the description of figure 1 in results - all the numbers are presented on the figure and the text only repeats them in very excessive form.

Author Response

Comments 1: Mutations j175-1 and j175-2 are loss of function mutations, Although the phenotype on figure 3 is strongly different from the phenotype of j175. The authors should explain this difference in phenotype and better explain why they relate the changes caused by j175-1 and j175-2  similar to the changes caused by j175. 

Response 1: Thank you for your suggestion. I forgot to add j175-1 and j175-2 as background of B73 inbred line in the Materials and Methods section (corrected), because the inbred lines are different and cannot be directly compared with j175 of RP125 background. But j175-1 and j175-2 only have different mutation sites, which can be compared and explained for differences. I have added this content in the results section.in compliance with “ with B73 inbred line background was obtained from maizeEMSDB.”and “As the allelic mutants of j175, j175-1 and j175-2, due to the higher position of the j175-1 mutation site, more structural domain functions are lost, resulting in a more severe reduction in agronomic traits for j175-2”.

Comments 2: Transcriptome Profiling of j175: I think the entire section is less informative because the connection between the gene and expression of other genes, and in your case it is hundreds of genes, is unclear. How it is connected to chloroplasts division? Functional classification group support this? Figures 4 B,C are  less informative because of the low number of genes. 

Response 2: Thank you for your suggestion. Due to the small amount of information, I have streamlined the text in the transcriptome section to make the description more concise. The transcriptome analysis results showed that some maize chloroplast division related genes were differentially expressed (Figure 4E), which was consistent with the qPCR results(Figure S16).

Comments 3: Reduce the description of figure 1 in results - all the numbers are presented on the figure and the text only repeats them in very excessive form.

Response 3: Thank you for your suggestion. I have appropriately streamlined the text describing Figure 1.For example, deleting certain parts “the chloroplasts in the leaf mesophyll cells of j175 mutant and wild-type plants. Normal mature mesophyll cells contained multiple chloroplasts of varying sizes” and “Additionally, no significant change was observed in the activity of SOD dismutase and Pro content”.

Round 2

Reviewer 3 Report

Comments and Suggestions for Authors

Authors replied to all questions